# Long Term Risks to Neonatal Health from Exposure to War—9 Years Long Survey of Reproductive Health and Contamination by Weapon-Delivered Heavy Metals in Gaza, Palestine

**DOI:** 10.3390/ijerph17072538

**Published:** 2020-04-08

**Authors:** Paola Manduca, Nabil Al Baraquni, Stefano Parodi

**Affiliations:** 1Department of Research, NWRG-onlus, 16123 Genoa, Italy; 2Department of Medicine, Islamic University of Gaza, Gaza 79702, Palestine; nbarqouni@gmail.com; 3Department of Epidemiology and Biostatistic Unit, IRCCS Istituto Giannina Gaslini, 16100 Genoa, Italy; stefanoparodi@gaslini.org

**Keywords:** surveillance at birth and prevalence of birth defects and preterm babies, heavy metal load in human cohorts, exposure to war and weapon-remnants, relevance of acute and chronic exposure to heavy metals in environment for reproductive health outcomes

## Abstract

*Introduction*: High levels of environmental contaminants with long term effects and teratogenic and carcinogenic potential, such as heavy metals, were introduced by weaponry in war areas in the last decades. Poorer reproductive health and increases in non-communicable diseases were reported after wars and are the suspected long term effects of contamination by stable war remnants. Although potentially affecting millions of people, this is still an understudied issue of public health. *Background*: Gaza, Palestine since 2006 has been an object of repeated severe military attacks that left heavy metals remnants in the environment, in wound tissues and that were assumed by the population. Retrospective studies showed a progressive increase in birth defects since the 2006 attacks. In 2011 we started surveillance at birth alongside analysis of the heavy metals load carried by pregnant women and their babies. *Methods*: We used protocols for birth registration which also document the extent of exposures to attacks, war remnants and to other environmental risks that allow comparison of 3 data sets—2011, 2016 and 2018–2019 (4000–6000 women in each set). By ICP/MS analysis we determined the content of 23 metals in mothers’ hair. Appropriate statistical analysis was performed. *Results*: Comparison of data in birth registers showed a major increase in the prevalence in birth defects and preterm babies between 2011 and 2016, respectively from 1.1 to 1.8% and from 1.1 to 7.9%, values remaining stable in 2019. Negative outcomes at birth in 2016 up to 2019 were associated with exposure of the mothers to the attacks in 2014 and/or to hot spots of heavy metals contamination. Metal loads since the attacks in 2014 were consistently high until 2018–2019 for barium, arsenic, cobalt, cadmium, chrome, vanadium and uranium, pointing to these metals as potential inducers for the increased prevalence of negative health outcomes at birth since 2016. *Conclusions*: Bodily accumulation of metals following exposure whilst residing in attacked buildings predispose women to negative birth outcomes. We do not know if the metals act in synergy. Trial for mitigation of the documented negative effects of high metal load on reproductive health, and ensuing perinatal deaths, could now be done in Gaza, based on this documentary record. High load of heavy metals may explain recent increases in non-communicable diseases and cancers at all ages in Gaza. Modern war’s legacy of diseases and deaths extends in time to populations and demands monitoring.

## 1. Introduction

In the last decade of the 20th century and continuing in the present one, intense military operations on wide geographic extensions have been taking place. From some of the affected countries, medical personnel reported an increase in negative birth outcomes [1,2,3,4,5] and in non-communicable diseases, including cancers, and similar news arrived from shooting grounds and retired military [6,7]. It is known that weaponry contributes to the deterioration of the environment causing acute events of pollution with cancerogenic and teratogenic substances; chronic contamination of the post-war areas by chemicals stable in time in the environment occurs and these are under scrutiny for the relevance for health of heavy metals.

Most heavy metals are carcinogens and teratogens or possible carcinogens [8]. They persist in the environment for a long, but unknown, length of time, causing chronic uptake by the living and are known to accumulate in animal bodies and plants. The molecular mechanism of the action of metals, has been studied mostly in animal models and in vitro—they can affect telomeric length [9], DNA methylation, alter the modification of histones and modulate gene expression and that of specific miRNAs [10], act as endocrine disruptors [11], alter transcription factors and receptors, affect the redox system and potentially impair the functionality of enzymes which contain metals. This has been confirmed by a limited number of studies in humans [9,12,13,14,15]. Metal loads that are effective in vivo are as yet undefined. Thus, heavy metals can affect reproductive health via epigenetic mechanisms. There is not a unique modality of action yet described exclusively for any of these metals while suggestions emerge that in a situation of multiple contamination they could act synergistically or antagonistically with each other. Heavy metals trans-pass the placenta during pregnancy, possibly with different mechanisms of passive and/or active transport, so that embryos and fetuses are exposed to them in utero [16] and they could be a primary agent of the increased risk of negative health outcomes at birth in the long term caused by war-remnants [17,18,19]. Environmentally available heavy metals determine their availability to the fetus [20] and another potential source of them could be the metals eventually already stored in the body of the mother; little is known about the release of the heavy metals accumulated in the body, for instance during pregnancy.

To date, we are not aware of any study of surveillance of reproductive health in any war area on a random cohort and in relation to documentation of the level of metal contamination of individuals over time. Correlation between documented individual exposure to military attacks, hair metals’ load and negative outcomes on reproductive health was reported in few observational studies [11,21,22,23]. Long term effects of prenatal exposure to metals were also reported on child and adult health [12,13,17]. Epigenetic regulation of prenatal development can be modified by the environment, and these changes contribute to growth, development, and disease risk of the new born and later in life [17,18,19].

The topic of long-term effects of weapon-derived metal contamination on reproductive health by weaponry and its quantitative impact in terms of death and diseases remains nonetheless substantially unexplored. Meanwhile exposures to metal delivering weaponry have been and are extensive [24,25,26,27,28], involving a very large number of people and could have effects on health for an unknown length of time in the future. 

After the military attack on Gaza in 2009, we sought to act quickly; previous observations suggesting long-term effects on health of war toxicants and teratogens, persisting in the environment, were not well documented and followed up. Since 200, we collected a number of environmental clues and data and we proved the presence of heavy metals in in the wounds produced by weaponry utilized in Gaza in 2006 and 2009 [27]; we showed the presence in the environment of metal war-remnants and their uptake by civilians a year after the 2009 military attacks [25,26], and the passage of metals in utero to newborns after a mother’s exposure to attacks [22]. We collected retrospective evidence of an increase in birth defects in Gaza since military airborne attacks started after Israelis retiring their settlements in the Gaza strip in 2005 [3,4]. The observational data suggested that studies of the health of exposed populations were urgently indicated. However, both traditional epidemiology and toxic assessments separately have limited potential in providing clinically crucial answers. We turned to prospective studies and undertook surveillance of birth outcomes, collection of information about the delivering women with an articulated ad hoc developed questionnaire designed to capture the date and location of exposure to contaminants. There was parallel assessment of contamination by heavy metals of newborns in 2011 [22,29], mothers and newborns in 2016 [30] and mothers in 2018 and 2019. We also assessed the load of heavy metals introduced in the environment and uptaken by women in 2015, following the major attacks in 2014 [24]. In 2011 we started implementing the tool of election, an articulated questionnaire adapted to the situation. Surveillance of birth records negative reproductive outcomes as an end point. Surveillance at birth is the most reliable tool, is cheaper, confers randomness and a relatively high number of cases in a short time, whilst privileging the detection of negative outcomes of pregnancy, and effects on prenatal development. Meanwhile, we used, along with surveillance, the measure of the level of contamination of mothers by heavy metals. We analyzed the load for 23 metals, and here report only on those with potential relevance and that were found in munitions used in Gaza [25,26] and found in wound tissue [27], and were associated with newborn birth defects or preterm birth in Gaza [22,30,31], and that have a higher overall level in all the cohorts at all times studied than in a reference population outside war areas. This allowed us to learn about the long-term effects of documented acute and potential chronical exposures with unprecedented details.

We here report the most recent results of this surveillance and compare them with those obtained over the last 9 years. We discuss the limits and the implications of these studies and the potential applicability they may have in a wider context.

## 2. Methods

### 2.1. Setting

Hospital based surveillance of births in the Shifa Maternity Hospital, the setting for more than 25% of births of the Gaza Strip. Catchment areas were the Gaza governorate (66% in 2011, and 72% in 2016 and 2017 and 2018–2019), the North governorate (15%–17%), the Middle governorate (3–5%) with less than 0.25% in other areas of the Strip of Gaza. Stability of residence since 2014 was also recorded. The procedure for obtaining the mothers’ informed consent for anonymous use of data and for hair sample collection has been described before [29,30].

### 2.2. Participants, Administration of Questionnaire, Classifications and Diagnosis

Participants were women at the point of their delivery; these were: 4049 mothers delivering a live baby after 28 weeks of gestation in 2011 (total deliveries including miscarriages and still born was 4173); 6104 mothers delivering a live baby after 28 weeks of gestation in 2016 (total deliveries including miscarriages and still born was 6185); 4765 mothers delivering a live baby after 28 weeks of gestation in 2018–2019 (total deliveries including miscarriages and still born was 4830). The questionnaire was administered in Arabic by a team of doctors in 2011 and by a team of licensed midwives, assisted by a doctor, in 2016, 2017 and 2018–2019. Data were collected uninterruptedly (24/24 hours and 7/7 days) in 2018–2019. The mother’s health and reproductive history, according to Eurocat [32] and classification of outcomes was according to CDC10 {33}. Briefly, Preterm (P) is any delivery from the 28th to the 36th week; low birth weight (LBW) at term are born at 37 or more weeks of gestation and of less than 2.5 kg; healthy babies born at term of 2.5 kg or more are normal birth weight (NBW). Diagnosis for birth defects included only major structural phenotypes, with systematic underestimation of the actual rate of some birth defects (for example minor cardiac and all metabolic diseases) and classification was according to ICD 10 [33]. Miscarriages are deliveries after 24 and before 28 weeks of gestation; there were not any earlier miscarriages. Total live born was the denominator for the calculation of prevalence values of birth defects (BD, preterm and LBW; total deliveries was the denominator to calculate prevalence for miscarriages and still born. In this report, as in previous registers, the prevalence of babies with BD includes preterm and small for weight (LBW), and the prevalence of babies of low weight (LBW) includes only normal babies. Prevalence values are expressed in percentages throughout. In Gaza, ultrasound services are available for pregnant women and interruption of pregnancy is subject to assessment of the risks for mother and child and performed in the maternity ward; interruption from 24 weeks on would be registered among miscarriages. No mother died. None of the women giving birth declined to respond and when, only occasionally, answers to some entries were missing we accounted for the missing numbers in the calculations. The use of the same methodology and procedures allows comparison of data from registers since 2011.

#### Ethical Approval

The study was conducted according to the indications of the WMA; The Palestinian Health Research Council, Helsinki Committee for Ethical Approval approved the study; the research tools and procedures were evaluated and accepted by an ad hoc Commission of the Ministry of Health in Gaza. 

Informed consent of the participants was obtained by the professional administering the questionnaire. The original files on paper with consent of the patients are now deposited at the Shifa Hospital and electronic copies at the Ministry of Health, Gaza, Palestine; documentation of receipt from the Human Resources Directorate of the Hospital is available on request.

### 2.3. Ad Hoc Auestionnaire

We have described the necessity and the modality of how to prepare “ad hoc” questionnaires [22,31,32] integrating the standard EUROCATs with more detailed queries about specific chronic diseases of the mother, reproductive history of the couple and presence in the extended family of cases of birth defects, education and employment of mother and father and nutritional status of the mother to analyze association of birth outcomes with known cofactors [30]. In addition to risk factors due to individual health and habits (smoke, medicines, drugs, nourishment), multiple queries were made about environmental exposures to potentially risky substances/circumstances in civilian life (including proximity of home to industry, agricultural land and to sites of unmanaged waste, use of pesticides and of household chemicals, and exposure to attacks and handling-reuse of goods from an attacked house). The questionnaire used in 2011 included questions related to military attacks in 2009; those used in 2016 and afterward related to exposures to successive military attacks and additional events, as infrastructural damages, recycling of ruins and rebuilding of infrastructures. Documentation about war damages and the weaponry used by the Israeli army, the timing and modalities of removal of ruins and reconstruction of infrastructures, were obtained from UNWRA, the UN mission for documentation and retrieval of spent weaponry, UNDEP, NGOs and from local environmental experts. These sources later published the observation they have anticipated for us [34,35,36,37,38,39,40,41,42]. More than 80 queries were used to evaluate predictive values for birth outcomes. The questionnaire used in 2018–2019 is in Appendix A.

### 2.4. Hair Samples and Metal Load Determination

Hair from mothers represents approximately the last 4–5 months of accumulation upon environmental exposure [43]. Mothers’ hair was analyzed in 2018–2019 for 17 mothers with a normal baby, 23 with BD and 24 with a preterm baby, born in sequence, as much as possible (some samples were insufficient for analytical purposes). The certified laboratory that tested metal hair load is the same as with previous tastings, using ICP/MS, in 3 runs and with the same protocol for analysis as previously described [22,24,30], to determine the concentration of 23 metals (Al, Fe, Mg, Mn, Ba, As, Cd, Co, Hg, Mo, Cr, Sr, Ti, U, V, Cs, Cu, Ni, Pb, Sn, W, Zn). We report only the load for 11 metals considered of relevance (Ba, As, Cd, Co, Hg, Mo, Cr, Sr, Ti, U, V). The unit of the concentration values reported is ppm (parts per million, i.e., mg/kg). A large international cohort from Europe was used as a reference [44]. We have analyzed a subgroup of the hair collected from women at delivery. In 2015 we could analyze a larger number of samples, while in 2016 and 2018–2019 we analyzed a similar number of hairs collected sequentially from mothers with normal, BD or preterm babies, being unable to test the whole cohort.

### 2.5. Statistical Methods

Descriptive statistics were reported as absolute frequencies and percentages for qualitative variables, and as medians with their related interquartile range (IQR) for quantitative ones. Differences in the prevalence of negative outcomes (low birth weight, birth defects and preterm birth) in the three considered periods were evaluated by a chi square test. A univariable and multivariable binary logistic regression model was employed to assess the association between each considered outcome and the following potential risk factors—mother and father age (continuous variables), parents consanguinity, residence area (North, Gaza, Middle, Kan Y and Rafah), gestational diabetes, new born gender, previous children with birth defects, mother and father education, working mother, birth defects in the extended family, parents’ use of fertilizer, residence near agricultural activities, residence close to sewage or garbage, residence under attack in 2014 and nearby houses under attack in 2014, and reuse of items from rubble. All analyses were restricted to singleton births (twin births were less than 2%). Multivariable models were built using a forward approach to reduce the overfitting bias [45] and only variables including at least ten individuals in each category were considered. Twenty three metals were analysed and here we report on those of potential relevance for health according to at least 2 of the 3 criteria—higher than in reference standard from outside war areas, with potential teratogenicity and carcinogenicity, and identified previously as weapons components (namely, barium, arsenic, cadmium, cobalt, chromium, mercury, molybdenum, selenium, strontium, titanium, uranium, vanadium). All analyses were carried out by using Stata for Windows statistical package (release 13.1, Stata Corporation, College Station, TX, USA). The conventional 0.05 threshold for the type I error was adopted to define statistical significance.

## 3. Results

Descriptive data of the register in 2018–2019 are reported in Table 1 for each outcome at birth and for most of the features for which recall was asked of the mother. A few features were omitted from this Table, because they had less than 0.5% prevalence in the whole cohort—maternal cigarette or shisha smoking, use of drugs or medication (3 entries), and maternal hypertension. Folic acid was also omitted since it was used by about 100% in each group of outcomes. 

The prevalence of negative outcomes at birth for the period of six months in 2018–2019 is reported in Table 2, compared with that in previous years. The number of deliveries with live born with 28 or more weeks of gestational age, registered in sequence were 4049 in 2011; 6104 in 2016; and 4765 in 2018–2019. The consistence of protocols and procedures for the registration in all years allows the comparison of data collected at different times. The prevalence of low birth at term, preterm and birth defects increased significantly since 2011 and remained not significantly different from 2016 to 2018–2019. Miscarriages and birth defects were calculated on total deliveries, including below 28 weeks of gestational age and dead babies, respectively 4173 in 2011, 6185 in 2016 and 4830 in 2018–2019; miscarriages decreased between 2011 and 2016 and did not change significantly after; still births were unchanged in the whole period 2011–2019. 

Table 2 shows that the most substantial increase in the prevalence of P, LBW and BD outcomes in the time span examined occurred after 2011 and before 2016, a period during which two major military attacks occurred (in 2012 and 2014) and major changes in the local environment accompanied by the extensive destruction of infrastructures, food dependence increased and when availability of all basic goods decreased, as widely independently documented by WHO, UNWRA, UNEP and other UN institutions. 

Appendix A shows the result of the logistic regression analysis, evaluating the association of the features in Table 1 with the type of birth outcome. Low weight at birth (LBW), as already reported for Gaza and worldwide, was associated with mother’s gestational diabetes, cesarean delivery and female sex of the child (*p* < 0.05). The lowest level of education (none or primary school), higher fish consumption (more than once a month), IVF conception and primiparity were also associated with the risk of LBW, but statistical significance was not reached. Baby’s gender, mother’s education, primiparity, diabetes and frequent fish consumption were confirmed as potential risk factors for LBW by multivariable logistic regression analysis (Table 3). 

BD outcome was associated with older maternal age, residence near factories and near a house hit by attacks in 2014. Father’s age and chronic disease and life in an agricultural setting were also associated with BD but the statistical significance was borderline. No association was found with cousin’s marriage or with previous birth defect in extended family or in the progeny of the couple, suggesting that most of the BD were sporadic and new events. By multivariable logistic regression analysis (Table 3) only residence near a house hit by attacks was significantly associated with the birth outcome.

Preterm birth was associated with male sex and vaginal delivery (an expected event for spontaneous preterm delivery), residence in the Middle governorate, with house hit by military attacks in 2014 and parents use of fertilizer; father’s education, previous BD in siblings or in the mother’s side of the family, mother being diabetic, and life in an agricultural setting had borderline statistical significance. Of these associations, gender, locality of residence, house hit in attacks in 2014 and reuse of rubble after attacks were validated by multivariable logistic regression analysis (Table 3). Univariable and multivariable analyses were conducted on singletons. Twins represented less than 3% in each year and none of them had birth defects. Personal habits were not found relevant in association with any outcome. 

The associations by multivariable logistic regression analysis for BD and preterm were confirmed when we analyzed only the subgroup of mothers whose residence remained unchanged since 2014 (not shown). 

Thus, recalled exposure to military attacks and related events of exposure to war-remnants were the environmental factors associated with birth defects and preterm birth outcomes. 

We analyzed for heavy metal content the hair collected at delivery from mothers each year. The metal load reflects the environmental exposure of the woman to toxicants/teratogens in the last 4-5 months. The metals shown in Table 4 and Table 5 include 12 of the 23 tested for, and include those potentially toxicant or teratogens (namely: barium, arsenic, cadmium, cobalt, mercury, molybdenum, chrome, strontium, titanium, uranium, vanadium) or were suspected to be cofactors in teratogenicity (selenium). All of them but selenium were identified previously in wound tissues or in ammunitions and craters from bombing in Gaza. We here present the results of the analysis of metal load for the mothers delivering in 2018–2019 in comparison with those of the analysis of mothers delivering in 2015 and 2016. The cohort for which metal analysis was done in 2015 was of 502 women (93% mothers of normal babies, 4% of BD, 3% of preterm; 68% of them were exposed to attacks and related events); in 2016 the cohort included 78 women (34.6% mothers of normal babies, 33.3% of BD and 32% of preterm; 47.5% of them were exposed to attacks and related events). In 2018–2019, 64 women were tested (27% mothers of normal babies, 37% BD and 36% preterm; 45% of them exposed to attacks and related events). Table 4 shows the comparison of the 95th percentile values in ppm of metal load in the cohorts for each year in comparison with a reference set of values from individuals residing in areas without military events (on the right in red); in blue are highlighted the values that are significantly higher in the Gaza cohorts compared to standards. Most of the metals toxic or teratogen (namely: barium, cadmium, cobalt, strontium, uranium, vanadium and chrome) have higher 95th percentile in Gaza mothers’ hair than in reference values at all times tested; arsenic was higher in 2016 and 2018, mercury previously higher; in 2018 it was not different than the reference, titanium was higher than the reference starting in 2016, while molybdenum and selenium were not higher than the reference at any time. None of the other 11 metals also tested, and not shown here, were present with 95th percentile load higher in Gaza than that of the reference sample outside war areas. The comparison of the median values of the load in metals in the cohorts of mothers (in Table 5) shows a trend of increase in time for the median values of arsenic, cadmium, cobalt, chrome, strontium, vanadium, titanium; steady levels of barium and uranium; for molybdenum and mercury the trend was to decrease and they were at the lowest observed values in 2018–2019. This comparison yields information consistent with the data in Table 4. Together, they point to high contamination of the environment by heavy metals war-remnants up to 2019 and point to those metals consistently in high or increasing amounts since 2015, and those that are not relevant for the birth outcomes, as Mo, Se throughout, and Hg only in 2018–2019.

In 2015 and 2016 the highest loads of mothers’ contamination were documented to be specifically associated with environmental events that we documented objectively, damage by direct military attacks and residence near hot spots of contaminants, respectively; in 2018–2019 we could only refer to the women’s recall since at this point war remnants had been obscured by reconstruction. The cohort tested for metal contaminants included in 2015 68% of women that recalled exposure to the 2014 attacks; in the 2016 cohort 40% of women recalled exposure to attacks and in 2018 these were 44%. We take these recalls at their face value; yet, since the recalls given in 2015 of exposure to attacks were photographically documented and 88% of the women in 2018–2019 had remained since these attacks in the same residence, it is possible that in 2016 and 2018–2019 there was under-recall. Anyhow, the fact that higher contamination for most metals than in 2015 was found in 2016 and 2018–2019 points to chronic uptake ongoing in 2016 and 2018–2019, of many metals teratogens that interfere with the progression of pregnancy to full term. It is worth noticing that the composition in terms of kind of birth outcome of the cohorts of the mothers analyzed for metal content and whose 95th percentile and median values are shown in Table 4 and Table 5, varied over the years—in 2015 the cohort included a lesser percentage of BD and preterm (total 7%) than in 2016 (total 65.3% BD and preterm) and in 2018 (73.5% BD and preterm). We do not know if and how much these differences contribute to the differences found in the overall level of contaminants among the cohorts, or if mothers with negative outcomes of progeny were more highly contaminated. Unfortunately, due to the low number of the available cases tested with negative birth outcomes, statistical analysis to compare metal loads between subgroups of birth outcomes was not reliable (not shown).

## 4. Discussion

In general, in war and post-war areas it is extremely difficult or impossible to carry out long term surveillance of reproductive health and of environmental factors, as the one presented here, and recommended by WHO among its SDG goals. This is because people may be displaced, because of the difficulties in the identification of an appropriate random cohort, in documenting exposures reliably (to attacks and other risk factors), in data collection and in identifying possible confounders. 

We started our survey of reproductive health in Gaza in the aftermath of a serious military operation in 2009, and, unexpectedly, we ended up working over a period plagued by two other major attacks in 2012 and 2014. Notwithstanding the difficulties and interruptions and the necessity of field recognition at each time for the major changes in the local environment that the attacks posed, the coexistence of objective conditions and participatory attitudes in the local situation made it possible to conduct a reliable study. 

As well as having to weather full-scale military attacks, Gaza has been under siege over the last 13 years during all these military attacks and it is a “specific” war and post-war situation [34,35,36,37,38,39,40,41,42]: There is virtually no immigration and stable residence in the housing for the participating population; appreciable underdevelopment or de-development of industry, agriculture and traffic, reduction of use of gasoline for electric generators, and of agriculture over the last 6 years which defined a situation with steady or decreased potential impact of chemical contamination by civilian activities. The environmental impact of the three major military attacks in 2008, 2012, 2014, have been well documented by local and international organizations. Timing of removal of ammunitions and ruins, changes in the handling of waste in the aftermaths of attacks were also documented by the local and the UN Environmental agencies, and these latter also provided maps of the strikes. The main Maternity Ward of al Shifa hospital provided unrestricted access to the delivery ward and we could reach a large (between 4000 and 6000 according to year) random cohort of women, a healthy constituency in reproductive age of all socioeconomic statuses. The medical authorities allowed, and the personnel in the Hospital supported, the study.

The period of surveillance lasted 9 years and bridged two intervening military attacks in 2012 and 2014, and the results in 2018–2019 are reported here in comparison with those of 2011 and 2016. There has been a steady higher risk of all negative outcomes in live births since 2011. The higher prevalence of babies of low weight at birth was confirmed to reflect mother’s health, food insecurity and lack of education, as already reported before in Gaza [30] and worldwide [46]. There was an association in 2018–2019, four and half years from the last major military attacks, between birth defects and preterm births with the mother’s recall of exposures to attacks in 2014, stable residence since in the same dwelling, and with activities thereafter associated with exposure to war-remnants (ruins removal, reuse of items, continuing residence in the housing hit or next to one hit). Regarding the data for 2018–2019, no significant risk factors from exposures in civil life (industry, chemicals for agriculture, work and householding) were found as confounders; residence near unmanaged waste, a risk factor in 2016, was no longer relevant in 2018–2019 and indeed proximity to unmanaged waste declined from more than 50% in 2016 to 9% in 2018–2019. Thus, the only environmental associations found for birth defect and preterm deliveries in 2018–2019 was exposure to military attacks, confirming the long-term effects these have on reproductive health. It is worth noticing that already in 2012 we presented evidence that contamination in utero of newborns was associated with their mother’s exposure to attacks that had occurred two years earlier [22]. Moreover, the cases of BD were in 2018–2019 largely progeny of couples without previous history of BD and of parents whose collaterals had no BD and were thus due to novel sporadic events. Also, the increment in the prevalence of BD since 2006 and registered already by 2011 was due to novel sporadic events [4,29]. Is not possible with the information available to discern whether these sporadic events associated with exposure to attacks occur through epigenetic changes or are due to an increased mutational rate in the germ line of exposed parents. 

Parallel collection of information documenting the evolution of a transforming environment was definitely a major task throughout the 9 years. One example of the complexities ensued from the delivery of military attacks during the period of surveillance arose from the observation in 2016 and 2018–2019 that women residing in the Middle governorate were more likely to have preterm babies than those residing in the Gaza governorate. We found out eventually that in 2014 there had been differences in the distribution of ground attacks/ air attacks in different areas of the Gaza strip which involved a different kind of weaponry and possibly of heavy metals components, information that may help give a rationale for these differences. 

Obtaining objective proof of women’s recall of direct exposures has been a priority in these studies to accompany epidemiological data and for their interpretation; the war events recalled by mothers were verified in the field in 2011 by reference to UNMAT maps, which plotted sites of explosion, and in 2016 through visits in person by our team. Objective checks had become impossible after rebuilding in 2018–2019 and we relied for the present study only on the recall of women and information of events from the local environmental agency EQA and UNDEP. The observed decline in recall of exposure to attacks after 2015 may be due to the difference in the size of the cohorts compared, or may be due to the phenomenon, elsewhere known, of refusing to be reminded of negative events. I is important to say that, whichever the reason, any eventual under evaluation of the percentage of women exposed to attacks in 2016 and 2018–2019 did not pose a quantitative challenge to the relevance of the association of exposure to attacks with the prevalence of BD and preterm deliveries, and would not impact the reported prevalence for negative outcomes or affect the interpretation of the trend of overall levels of contamination presented. 

What we measured by the kind of sampling and methodology used is the most recent exposure of women to metals in their environment. Contamination in 2018–2019 by the 12 metals here shown was unabated since 2015 and at four and a half years from the last attacks—this was similar to that measured in 2015 and 2016 for barium and uranium and was higher than these for arsenic, cadmium, cobalt, chrome, strontium and vanadium [24]. The load for these metals was always higher than outside war areas. All the metals reported here trespass the placenta [11,12,16,22,24].

The data documents the continuing availability for uptake by women of teratogens and carcinogenic metals potentially interfering with the outcome of pregnancy since 2015; they also confirm our previous suggestion that the removal of bombed debris in 2016 may have determined a wide spread of metal- rich materials embedded in ruins. We have no fact-grounded suggestions to interpret why mercury and molybdenum uptake by women decreased in 2018-2019; after 2015 selenium and molybdenum were steadily at reference level, while mercury decreased. The stability of most toxicant and teratogens over more than 4 years from the main military attacks in Gaza is in concordance with results obtained in another war aftermath, in Fallujah, Iraq, where 5 years after major military attacks the level of heavy metal contamination of women, newborn and children was generally very high ([21], Manduca, unpublished data). Our data further confirm that availability at risky concentrations of heavy metals is a long-term legacy of the use of the modern weaponry. In 2018–2019 the women who were more likely to deliver a birth defect or a preterm baby were those that recalled exposure to past attacks and related events, and with stable residence in the site of attacks, that is, who have had both acute and long continual chronic exposures. These exposures may compound in time. There were no confounding factors linked to familiarity of the outcome or to consanguinity of the parents or to any other known factor for these outcomes at birth. This association of negative outcomes with past and long chronic exposure to metal delivering war-remnants is somehow countering a working hypothesis that assumes that the chance of new negative events at birth might depend only or primarily from the level of most recent contamination during the present pregnancy. We suggest instead that there may be an additive impact of a past acute exposure and long term chronic and nearest exposures, mediated by the accumulation of heavy metals in the body, and this metals storage contributes to the achievement of a threshold relevant to promoting negative outcomes at birth in pregnancies at a later time. This suggestion amounts to opening a question for investigation. Obtaining proof of its soundness will require studies on animals and more invasive procedures with molecular methodologies of investigation.

The relevance of identifying and assessing the impact of environmental factors during pregnancy is due to the various implications that derive from this information. Besides showing the direct effect on reproductive health, from which ensue implications about neonatal mortality, provision of needs for care giving at birth and in infancy to fragile children, which we discuss here. Later effects on infant and child health may also occur when epigenetic modifiers act at the time of greatest plasticity during embryo and fetal development [13]. Association of prenatal exposures with high levels of heavy metals with the development of diseases at later times, in coherence with the developmental origin of diseases, has been reported in various experimental and different environmental contexts in peace and war [5,13,17]. Lead, mercury, cadmium, arsenic and vanadium [12] uptake by mothers during pregnancy were implied in the impairment of DNA methylation in newborns and in later development in childhood [13]. We do not include lead in this report, even if in high load it is implied in other contexts in determining oxidative stress [47], because lead levels measured in Gaza were at all times lower than in reference outside the war area, consistent with the limitation of the usage of gasoline for all purposes and decline of manufacturers, due to post 2014 war shortages and siege. In Gaza, two years after the extensive use of white phosphorus ammunitions containing mercury in 2009, there was an association of birth defects with the highest level of mercury in newborn hair [22]. In the following military attacks in 2012 and 2014 no use of white phosphorus ammunitions was reported. Mercury was still detected in high amounts in the hair of mothers with a higher frequency of negative birth outcomes in 2016, but it had reached much lower levels by 2018–2019. This may suggest that it has somehow been eliminated in the environment 10 years after its introduction.

Cadmium was reported to be teratogenic in humans, and arsenic is an epigenetic modifier associated with possible long-term health risks [14,15]. Cadmium and arsenic loads in mothers increased over time, but they were not associated with a higher risk of negative outcomes at birth in 2016. Vanadium, a potential teratogen [12] whose load in mothers’ hair increased in time, was in 2016 found to be associated with a higher frequency of negative outcomes at birth. Uranium and barium loads were higher than those out of war areas, consistent with their use in many types of ammunitions, and steady. Both are known effectors of reproductive health [14,15]. In 2011 a high load of barium was detected in newborn preterm and in 2016 a higher frequency of birth defects and preterm deliveries was documented for mothers with a high barium load. The load in mothers of cobalt, a known carcinogen, has increased since 2015. Higher levels of cobalt in 2016 were found in mothers more likely to have a negative outcome for the newborn. Molybdenum and titanium loads had no association with negative outcomes in 2016 and were always in the range of values of reference for outside war areas. Strontium, with known potential as a toxicant and teratogen, was in the range of reference and, although levels have increased since 2015, it may not be of relevance per se to the increase in the prevalence of negative birth outcomes.

Few reports are available from other situations where, as in Gaza, multiple metal loads in excess have been absorbed at the same time, and where the potential for combined and cumulative effects upon reproductive outcomes have been studied [21]. 

Disturbances of the emotional and psychological development in toddlers of the cohort studied in 2015 in Gaza were reported, associated with high levels of chrome and uranium as in utero contaminants [48]. 

It is important to learn which heavy metals are constantly involved in an association with negative events at birth. Surely, tracing the effects on health of the exposure to heavy metals poses great challenges for the lack of knowledge about many important issues: (1) mechanisms of heavy metals’ interference with biological processes (stochastic/deterministic/threshold levels), (2) interactions between metals in the body in presence of multiple elements (synergy/inhibition, epistatic effects), (3) storage and recirculation/repositioning in specific metabolic circumstances, as during pregnancy, risky concentrations for the embryo and fetus, (4) identity of their molecular targets. Also, individual metal longevity in the environment might vary, and here we show that for mercury, in the absence of repletion by further attacks with white phosphorus ammunitions, it may have extended from 2009 to 2016 and declined by 2018. 

We were not able to implement, although we had planned these since 2014, studies on the eventual changes of the epigenome paralleling exposure to war events and to individuals’ metal load due to the lack of energy for cryopreservation of the needed fresh bioptic samples, and of supplies for the laboratory in Gaza; yet this approach may be of support in developing protocols for remediation. Another aspect of relevance of these studies which we could not tackle for the above mentioned scarcities, is the role of environmental contamination by heavy metals in view of the cross resistance of bacteria to metals and to antibiotics [49]. Increase in antibiotic resistant strains of endemic or pathologic bacteria has been signaled since 2014 in Gaza.

About 50% of perinatal deaths in Gaza are due to preterm birth and BD; the rise of prevalence from a total of 2.2 % in 2011 to 9.5% since 2016, and ongoing, of babies born with these conditions associated with exposure to war, gives an estimate for the whole Gaza strip (60,000 newborn per year) of an additional few thousand newborn deaths/year. An increase in perinatal mortality since 2013, and its stabilization by 2018 have also been documented [50,51]. The increase in numbers of babies that survive with precarious health that are more fragile or in need of medical care add a toll to the families and to the hugely overstressed health system. In addition, it can be hypothesized that other negative events of health could occur in the development of babies as consequence of in utero exposure to heavy metals. In summary, besides the few answers, our data contribute to raising plenty of questions and point to the fact that different methodological approaches in research are needed.

The facts presented point to the more urgent issue to tackle, that of protection of lives. One main drive of our work has been to provide evidence that is useful to develop prevention, possibly without collateral effects and of low cost, taking an example from that widely implemented with folic acid. Having verified the situation with long surveillance, we are in the condition to design a trial for such prevention.

## 5. Conclusions

The results of the surveillance of health at birth in the last 9 years in Gaza confirmed that exposure to military attacks was the main event associated with the increase of birth defects and preterm births since 2011, in the absence of other confounders. Parallel monitoring of women’s contamination by heavy metals showed that, four and a half years past the last major attacks in 2014, the level of many risky heavy metals continued to be high or higher compared to their level in the immediate aftermath, with the exception of mercury, which decreased. Accumulation of high levels of weapon-derived metals in the bodies of women may play a role in predisposing them to reproductive morbidities for years to come. Cumulative exposure to past attacks and chronic exposures may determine accumulation above a threshold, possibly compounded by individual susceptibility, which impairs the biological events of morphogenesis in utero and also those involved in the mother’s capability to bring a pregnancy to term. This study also contributes to underlining that seeking “environmental determinants of health” cannot be achieved appropriately using only the “one for all” questionnaire for birth registration recommended by the WHO. Only information sought on the basis of each specific context, including historical information about environmental changes and extended family health can lead to interpretation of eventual changes in the incidence of negative birth outcomes. This would guide the choice of specific objective tests for environmental effectors, and possibly lead to proposals for remediation. The methods employed in this work may be of value also for studies of reproductive health in other complex situations of war aftermath, or in industrial disasters. This is the first study where attention was devoted to documenting the relationship between a changing environment, uptake of heavy metals and birth outcomes, adopting an evolving long-term vision. We also identified which heavy metals remained bioavailable and could be responsible, alone or in synergy, for the negative effects on reproductive health.

### Limits

The major limit of the study is that the number of analytical measures were not sufficient for testing significance among subgroups for outcomes at birth. We hope to acquire soon enough funds to be able to test in the future more samples from our bank of hair and to increase the number of metal load measures to address the issue of statistical power. 

## Figures and Tables

**Table 1 ijerph-17-02538-t001:** Parents’ characteristics and exposures, reproductive history, and delivery complications. Year 2018–2019.

	ALL (N = 4765)		Birth Defects (N=78)		Preterm Birth (N=360)		Low Birth Weight (N = 142)	
Features	N	%	N	%	N	%	N	%
Parents characteristics								
Mother age (year)								
<20	492	10.4	8	10.5	40	11.2	14	10.0
20-29	3158	66.7	44	57.9	235	65.6	88	62.9
30-39	984	20.8	18	23.7	79	22.1	34	24.3
≥ 40	98	2.1	6	7.9	4	1.1	4	2.9
Missing	(33)	(0.7)	(2)	(2.6)	(2)	(0.6)	(2)	(1.4)
Mother education								
No school	106	2.2	0	0.0	9	2.5	8	5.6
Primary	678	14.3	16	20.8	56	15.6	20	14.1
Secondary or more	3974	83.5	61	79.2	293	81.8	114	80.3
Missing	(7)	(0.2)	(1)	(1.3)	(2)	(0.6)	(0)	(0.0)
Working mother								
No	4513	95.1	71	93.4	337	93.9	131	92.3
Yes	231	4.9	5	6.6	22	6.1	11	7.8
Missing	(21)	(0.4)	(2)	(2.6)	(1)	(0.3)	(0)	(0.0)
Primiparity								
No	3258	68.4	52	66.7	236	66.3	91	64.1
Yes	1500	31.5	26	33.3	120	33.7	51	35.9
Missing	(7)	(0.2)	(0)	(0.0)	(4)	(1.1)	(0)	(0.0)
Previous children with birth defects								
No	4721	99.2	77	98.7	352	98.6	142	100
Yes	36	0.8	1	1.3	5	1.4	0	0.0
Missing	(8)	(0.2)	(0)	(0.0)	(3)	(0.8)	(0)	(0.0)
Birth defects in the mother’s family								
No	4624	97.3	73	96.1	341	95.3	141	99.3
Yes	130	2.7	3	3.9	17	4.7	1	0.7
Missing	(11)	(0.2)	(2)	(2.6)	(2)	(0.6)	(0)	(0.0)
Father age (year)								
<20	13	0.3	1	1.3	0	0.0	0	0.0
20-29	2599	55.0	36	48.0	200	55.9	76	54.3
30-39	1687	35.7	25	33.3	131	36.6	54	38.6
≥ 40	423	9.0	13	17.3	27	7.5	10	7.1
Missing	(43)	(0.9)	(3)	(3.9)	(2)	(0.6)	(2)	(1.4)
Father education								
No school	243	5.1	2	2.7	22	6.2	9	6.3
Primary	1005	21.1	19	25.3	81	22.6	34	23.9
Secondary or more	3505	73.7	54	72.0	255	71.2	99	69.7
Missing	(12)	(0.3)	(3)	(3.9)	(2)	(0.6)	(0)	(0.0)
Father chronic diseases								
No	4730	99.3	73	97.3	356	99.4	142	100
Yes	25	0.5	2	2.7	2	0.6	0	0.0
Missing	(10)	(0.2)	(3)	(3.9)	(2)	(0.6)	(0)	(0.0)
Birth defects in the father family								
No	4624	97.3	71	94.7	346	96.7	138	97.2
Yes	129	2.7	4	5.3	12	3.4	4	2.8
Missing	(12)	(0.3)	(3)	(3.9)	(2)	(0.6)	(0)	(0.0)
Parents consanguinity								
No	2866	60.2	38	50.7	201	56.3	85	59.9
Cousin	694	14.6	13	17.3	53	14.9	21	14.8
Relative	1180	24.9	24	32.0	103	28.9	36	25.4
Missing	(25)	(0.5)	(3)	(3.9)	(3)	(0.8)	(0)	(0.0)
Delivery features								
Infant sex								
Male	2383	51.7	36	47.4	177	58.2	41	37.3
Female	2224	48.3	40	52.6	127	41.8	69	62.7
Missing	(158)	(3.3)	(2)	(2.6)	(56)	(15.6)	(32)	(22.5)
Twin birth								
No	4610	96.9	77	98.7	305	85.0	113	79.6
Yes	149	3.1	1	1.3	54	15.0	29	20.4
Missing	(6)	(0.1)	(0)	(0.0)	(1)	(0.3)	(0)	(0.0)
Gestational diabetes								
No	4515	94.8	74	97.4	335	93.6	130	91.6
Yes	241	5.1	2	2.6	23	6.4	12	8.4
Missing	(9)	(0.2)	(2)	(2.6)	(2)	(0.6)	(0)	(0.0)
Type of delivery								
Vaginal	3702	77.8	56	71.8	236	65.6	81	57.0
Cesarean	1054	22.2	22	28.2	124	34.4	61	43.0
Missing	(9)	(0.2)	(0)	(0.0)	(0)	(0.0)	(0)	(0.0)
In vitro fertilization								
No	4703	98.7	77	98.7	345	95.8	132	93.0
Yes	62	1.3	1	1.3	15	4.2	10	7.0
Missing	(0)	(0.0)	(0)	(0.0)	(0)	(0.0)	(0)	(0.0)
Residence characteristics								
Residence area								
North	701	14.7	12	15.4	56	15.6	16	11.3
Gaza	3906	82.0	62	79.5	267	74.2	122	85.9
Middle	149	3.1	4	5.1	34	9.4	4	2.8
KanYounes	5	0.1	0	0.0	1	0.28	0	0.0
Rafah	4	0.1	0	0.0	2	0.56	0	0.0
Missing	(0)	(0.0)	(0)	(0.0)	(0)	(0.0)	(0)	(0.0)
House hit in 2014								
No	3691	79.4	58	80.6	257	73.9	111	79.3
Yes	956	20.6	14	19.4	91	26.2	29	20.7
Missing	(118)	(2.5)	(6)	(7.7)	(12)	(3.3)	(2)	(1.4)
Nearby house hit in 2014								
No	3753	79.7	48	65.8	273	77.1	119	83.8
Yes	957	20.3	25	34.3	81	22.9	23	16.2
Missing	(55)	(1.2)	(5)	(6.4)	(6)	(1.7)	(0)	(0.0)
Rubble use after 2014								
No	1791	37.7	25	33.3	150	41.9	53	37.3
Yes	2959	62.3	50	66.7	208	58.1	89	62.7
Missing	(15)	(0.3)	(3)	(3.9)	(2)	(0.6)	(0)	(0.0)
House near a sewage								
No	4372	92.1	67	89.3	324	90.5	134	94.4
Yes	377	7.9	8	10.7	34	9.5	8	5.6
Missing	(16)	(0.3)	(3)	(3.9)	(2)	(0.6)	(0)	(0.0)
House near a garbage								
No	4557	96.0	74	98.7	348	97.2	137	96.5
Yes	192	4.0	1	1.3	10	2.8	5	3.5
Missing	(16)	(0.3)	(3)	(3.9)	(2)	(0.6)	(0)	(0.0)
House near agricultural activities								
No	3792	79.8	66	88.0	273	76.3	117	82.4
Yes	959	20.2	9	12.0	85	23.7	25	17.6
Missing	(14)	(0.3)	(3)	(3.9)	(2)	(0.6)	0	0.0
House near factories								
No	4625	97.4	67	89.3	351	98.0	140	98.5
Yes	126	2.6	8	10.7	7	2.0	2	1.4
Missing	(14)	(0.3)	(3)	(3.9)	(2)	(0.6)	(0)	(0.0)
Mother still resident in the same area after 2014								
No	304	6.4	7	9.3	28	7.8	6	4.2
Yes	4448	93.6	68	90.7	330	92.2	136	95.8
Missing	(13)	(0.3)	(3)	(3.9)	(2)	(0.6)	(0)	(0.0)
Lifestyle exposures								
Folic acid during pregnancy								
No	12	0.3	0	0.0	0	0.0	0	0.0
Yes	4516	99.7	72	100	347	100	134	100
Missing	(237)	(5.0)	(6)	(7.7)	13	(3.6)	(8)	(5.6)
Mother fish consumption								
No	698	14.7	13	17.3	60	16.8	11	7.8
Yes	4050	85.3	62	82.7	298	83.2	131	92.3
Missing	(17)	(0.4)	(3)	(3.9)	(2)	(0.6)	(0)	(0.0)
Parents use of fertilizers								
No	4071	85.4	61	82.4	294	82.1	119	83.8
Yes	671	14.1	13	17.6	64	17.9	23	16.2
Missing	(23)	(0.5)	(4)	(5.1)	(2)	(0.6)	(0)	(0.0)

*Notes*: Table 1 Descriptive data of the register 2018–2019. N = number of subjects; % = percentages calculated on valid data only.

**Table 2 ijerph-17-02538-t002:** Prevalence of outcomes at birth in the years 2011, 2016 and 2018–2019.

	2011		2016		2018–2019					
Features	No	Yes	No	Yes	No	Yes	Chi2 Overall (2 d.f.)	*p*-Value Overall	*p*-Value 2011 vs 2018–2019	*p*-Value 2016 vs 2018–2019
Preterm *	4004	45	5622	482	4405	360	232.8	<0.001	<0.001	0.509
	98.9%	1.1%	92.1%	7.9%	92.4%	7.6%				
LBW *	4021	28	5945	159	4623	142	60.8	<0.001	<0.001	0.237
	99.3%	0.69%	97.4%	2.6%	97.0%	3.0%				
BD *	4005	44	5996	108	4687	78	7.89	0.019	0.028	0.597
	98.9%	1.1%	98.2%	1.8%	98.1%	1.9%				
Miscarriage **	4079	94	6144	41	4809	21	86.7	<0.001	<0.001	0.112
	97.7%	2.3%	99.3%	0.66%	99.6%	0.41%				
Still Born **	4143	30	6145	40	4786	44	2.61	0.271	0.319	0.114
	99.3%	0.72%	99.4%	0.65%	99.1%	0.95%				

Chi2 = chi squared test for heterogeneity, d.f. degrees of freedom. LBW = Low Birth Weight; BD = Birth Defects. Miscarriages include 3 babies born at birth. Still born are babies of more than 28 weeks gestation.* Total live born = 4765, ** total delivered babies = 4830.

**Table 3 ijerph-17-02538-t003:** Association between singleton babies with birth defects, preterm and of low birth weight at term and babies’ and mothers’ characteristics, evaluated by multivariable logistic regression analysis *.

Features	OR	95% CI	*P*
*Birth Defects ***			
Nearby house hit in 2014			0.003
No	Ref	-	
Yes	2.1	1.3–3.4	
*Preterm birth*			
Residence Area ***			<0.001
North	Ref	-	
Gaza	0.79	0.56–1.1	
Middle	3.5	2.1–5.8	
House hit in 2014			0.003
No	Ref	-	
Yes	1.5	1.1–2.0	
New born gender			0.019
Male	Ref	-	
Female	0.75	0.59–0.95	
Rubble use after 2014			0.019
No	Ref	-	
Yes	0.74	0.57–0.95	
*Low birth weight*			
New born gender			0.003
Male	Ref	-	
Female	1.8	1.2–2.7	
Mother education: secondary school or more			0.004
No	Ref	-	
Yes	0.51	0.32–0.81	
Mother fish consumption			0.020
Less than once a month	Ref	-	
More than once a month	2.3	1.1–4.7	
Primiparity			0.037
No	Ref	-	
Yes	1.5	1.0–2.3	
Gestational diabetes			0.053
No	Ref	-	
Yes	1.9	0.99–3.8	

OR = Odds Ratio; 95%CI: 95% confidence intervals. * Forward selection. Only predictors with at least 10 observation by group have been analyzed; twin births were excluded from the analyses. ** Only one predictor selected (the multivariable model is equivalent to the univariable one). *** Kan Yunes and Rafah areas excluded for insufficient sample size.

**Table 4 ijerph-17-02538-t004:** 95th percentile values of metal concentration in ppm in the hair of Gaza mothers at delivery and in reference cohort outside war areas.

	2015			2016			2018–2019			
Metal	N	95th pct	95% CI	N	95th pct	95% CI	N	95th pct	95% CI *	Ref.
**Ba**	502	29.69	24.0–49.1	78	44.6	31.9–1870.9	64	29.9	17.6–1600.0	<4.64
**As**	502	0.234	0.20–0.27	78	0.463	0.193–30.6	64	0.456	0.369–11.6	<0.2
**Cd**	502	0.24	0.2–0.3	78	0.66	0.43–1.2	64	0.97	0.50–2.6	<0.2
**Co**	502	0.57	0.37–0.76	78	1.68	0.99–6.8	64	13.1	1.7 -33.0	0.01–0.30
**Cr**	502	2.91	2.53–3.3	78	3.73	2.7–12.1	64	4.4	3.3–7.4	0.01–0.20
**Hg**	502	1.63	1.1–4.8	78	2.16	0.94–20.5	64	0.76	0.26–22.4	<0.6
**Mo**	502	0.26	0.21–0.32	78	0.22	0.12–0.41	64	0.08	0.07–0.214	0.03–1.00
**Se**	502	0.88	0.86–0.95	78	0.75	0.51–9.1	64	1.16	0.88–26.1	0.40–1.70
**Sr**	502	136	122.3–160.2	78	201	109.8–436.3	64	166	124.3–418.0	0.65 -6.90
**Ti**	502	0.82	0.73-1.0	78	2.75	1.5–5.6	64	3.1	1.7–6.1	<1.50
**U**	502	0.53	0.46-0.68	78	0.41	0.29–0.93	64	0.54	0.36–1.7	<0.10
**V**	502	1.4	1.26-1.56	78	1.8	1.3–6.5	64	1.4	1.1–2.0	0.01–0.20

N, number of tested samples. 95th pct = 95th percentile; 95%CI: 95% confidence intervals of 95th pct. Ref. = Reference values. * For the 2018–2019 estimates, the upper confidence limit held at maximum of sample due to the small sample size. Highlighted in blue, are the values significantly higher than those in the reference samples from area not involved in military attacks, from Western countries.

**Table 5 ijerph-17-02538-t005:** Comparison of the median metal concentration in hair of the cohort of mothers by period.

	2015 (N = 502)		2016 (N = 78)		2018–2019 (N = 64)		*p*
Metal	Median	IQR	Median	IQR	Median	IQR	
**Ba**	4.8	2.5–10.3	5.6	3.7–12.2	6.9	3.2–11.3	0.056
**As**	0.07	0.04–0.12	0.05	0.03–0.09	0.16	0.04–0.25	<0.001
**Cd**	0.05	0.02–0.09	0.15	0.07–0.23	0.14	0.09–0.23	<0.001
**Co**	0.05	0.02–0.13	0.11	0.04–0.20	0.20	0.07–0.63	<0.001
**Cr**	0.67	0.33–1.2	1.2	0.69–1.9	1.2	0.70–2.2	<0.001
**Hg**	0.19	0.10–0.35	0.29	0.13–0.44	0.01	0.0–0.08	<0.001
**Mo**	0.06	0.04–0.11	0.06	0.03–0.08	0.02	0.0–0.04	<0.001
**Se**	0.65	0.54–0.74	0.33	0.20–0.45	0.42	0.28–0.58	<0.001
**Sr**	49.2	32.0–75.0	51.1	36.2–76.1	66.5	44.4–89.3	0.018
**Ti**	0.26	0.16–0.41	0.59	0.28–0.92	0.34	0.04–0.69	<0.001
**U**	0.15	0.09–0.27	0.12	0.07–0.20	0.15	0.06–0.27	0.041
**V**	0.43	0.19–0.74	0.70	0.41–1.1	0.64	0.37–0.98	<0.001

IQR = Interquartile Range; *p* = *p*-value for the comparison by period (Kruskal-Wallis test).

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
