# Peer review of "Long Term Risks to Neonatal Health from Exposure to War—9 Years Long Survey of Reproductive Health and Contamination by Weapon-Delivered Heavy Metals in Gaza, Palestine"

_ijerph, 2020, doi:10.3390/ijerph17072538_

Round 1

Reviewer 1 Report

Dear Authors,

the topic of your article is very interesting and I wish to congratulate with you for addressing this topic. However, you must revise English and several parts are not clear for the potential readers. You must do an effort to improve and re-write some part of the manuscript.

I have some comments to improve the understandability for readers and the general quality of your manuscript.

First of all, I think that the manuscript needs an extensive revision of English.

Also, the abstract is too large. You should reduce it and you must be more concise. You must insert some fundamental information, like as the sample size. Also, you may revise the keywords.

lines 115-116. You must indicate the number of tables. You cannot cite them without writing numbers.

line 118. You cannot write "described in ref. 32.". You must write it like the other references [...].

Table 2 should be inserted before Table 1. In fact, first of all you must show the characteristics of the subjects involved in the study.

It is not clear how many subjects were included in the study. In table 1 you wrote N=4765, while in Table 2 you wrote other numbers. Please, you should clarify the sample size in the text. You should insert it also in the abstract.

Line 152. 2.4. Hair samples and metal load determination
I do not understand why only few mothers have done the hair sample. I am not able to understand the total number of hair samples. The text does not seem consistent whit date reported in table 4A. Please, you should clarify this part.

Reviewer 2 Report

Thanks for the opportunity to review this wonderful paper: My review can be found below:

Abstract:

Line 12 –live not leave…. Change to… “{People live in environments that are gradually becoming for toxic to human health due to the use of chemicals such as heavy metals which persist in the environment. “

Line 15 makes little sense.  “Why not say, the effects of war environments on reproductive health is an understudied issue of public health significance”.

There are more in the abstract. Please revise carefully…

Introduction:

The introduction is good but misses some key things.

You should also speak about sources of lead exposure:

Obeng-Gyasi, E. (2019). Sources of lead exposure in various countries. Reviews on environmental health, 34(1), 25-34.

Then speak about lead exposure in military environments:

Greenberg, N., Frimer, R., Meyer, R., Derazne, E., & Chodick, G. (2016). Lead exposure in military outdoor firing ranges. Military medicine, 181(9), 1121-1126.

Obeng-Gyasi, E., & Obeng-Gyasi, B. (2018). Blood Pressure and Oxidative Stress among US Adults Exposed to Lead in Military Environments—A Preliminary Study. Diseases, 6(4), 97.

And touch on how exposure affects the exposed during their lifetime: 

Reuben, A., Caspi, A., Belsky, D.W., Broadbent, J., Harrington, H., Sugden, K., Houts, R.M., Ramrakha, S., Poulton, R. and Moffitt, T.E., 2017. Association of childhood blood lead levels with cognitive function and socioeconomic status at age 38 years and with IQ change and socioeconomic mobility between childhood and adulthood. Jama, 317(12), pp.1244-1251.

Obeng-Gyasi, E., 2018. Lead Exposure and Oxidative Stress-A Life Course Approach in US Adults. Toxics, 6(3).

Methods:

 Well done:

Results: Please fix table 4A. Not well done at all.

Discussion: Please compare and contrast the results of this study with other studies. More of that is needed to put the results in the proper context.  

Round 2

Reviewer 1 Report

Dear Authors,

the manuscript is significantly improved thanks to your changes.

I think that it can be published on the IJERPH.

Best regards

Author Response

no further request to answer

Reviewer 2 Report

Significant improvement. Well done!

Author Response

No further requests to answer